

# An autonomous mixed data oversampling method for AIOT-based churn recognition and personalized recommendations using behavioral segmentation

Ghulam Fatima[1,*], Salabat Khan[1,2,*], Farhan Aadil[1], Do Hyuen Kim[3], Ghada Atteia[4] and Maali Alabdulhafith[4]

[1] Department of Computer Science, Comsats University Islamabad, Attock Campus Pakistan, Attock, Punjab, Pakistan
[2] Big Data Research Center, Jeju National University, Jeju, Korea
[3] Department of Computer Engineering, Jeju National University, Jeju Special Self-Governing Province, South Korea
[4] Department of Information Technology, College of Computer and Information Sciences, Princess Nourah bint Abdulrahman University, Riyadh, Saudi Arabia
* These authors contributed equally to this work.

Corresponding authors
Salabat Khan,
salabat.khan@ciit-attock.edu.pk
Maali Alabdulhafith,
mialabdulhafith@pnu.edu.sa

## ABSTRACT

The telecom sector is currently undergoing a digital transformation by integrating artificial intelligence (AI) and Internet of Things (IoT) technologies. Customer retention in this context relies on the application of autonomous AI methods for analyzing IoT device data patterns in relation to the offered service packages. One significant challenge in existing studies is treating churn recognition and customer segmentation as separate tasks, which diminishes overall system accuracy. This study introduces an innovative approach by leveraging a unified customer analytics platform that treats churn recognition and segmentation as a bi-level optimization problem. The proposed framework includes an Auto Machine Learning (AutoML) oversampling method, effectively handling three mixed datasets of customer churn features while addressing imbalanced-class distribution issues. To enhance performance, the study utilizes the strength of oversampling methods like synthetic minority oversampling technique for nominal and continuous features (SMOTE-NC) and synthetic minority oversampling with encoded nominal and continuous features (SMOTE-ENC). Performance evaluation, using 10-fold cross-validation, measures accuracy and F1-score. Simulation results demonstrate that the proposed strategy, particularly Random Forest (RF) with SMOTE-NC, outperforms standard methods with SMOTE. It achieves accuracy rates of 79.24%, 94.54%, and 69.57%, and F1-scores of 65.25%, 81.87%, and 45.62% for the IBM, Kaggle Telco and Cell2Cell datasets, respectively. The proposed method autonomously determines the number and density of clusters. Factor analysis employing Bayesian logistic regression identifies influential factors for accurate customer segmentation. Furthermore, the study segments consumers behaviorally and generates targeted recommendations for personalized service packages, benefiting decision-makers.

# INTRODUCTION

Customer churn data analytics is particularly valuable in the industries where the turnover rate is high accompanied with diverse customer service packages. Customer churn recognition and segmentation are the two integral parts for the analytical analysis in the business's growth. As a case study, we consider telecom sector which has become one of the essential parts to the evolution and expansion of modern industrialized state. The telecom sector is experiencing a digital transformation with the integration of artificial intelligence (AI) and Internet of Things (IoT) technologies. This integration has resulted in a large volume of data being generated from various connected devices, such as smartphones, routers, and wearables. The main takeoff boosters of telecom industry are the increased number of service providers, technical advancements, and entrance of many competitors in the field (*Gerpott, Rams & Schindler, 2001*). The arrival of 5G technology have generated much more opportunities for Telco business but it also increased the rate of customer preferential switching at a fast pace. Such huge business chances also lead to aggressive competition in the telecommunication companies, along with high churning rate. It is critical for the telecom companies to come up with personalized marketing policies (using customer analytics) in order to avoid the customer's churn for boosting the company's revenue. In the telco industry, it is not a suitable procedure to attract new customers while allowing existing customers to churn because the cost of the retention of the existing customers is much lower as compared to the cost of enrolling the new ones (*Esteves & Mendes-Moreira, 2016*).

Various types of methods are considered for controlling the customer churn and can be broadly categorized as reactive and proactive methods (*Tsai & Lu, 2009*). The proposed method is a proactive approach which identifies customer churn in advance and make personalized recommendation plans for decision makers. Our customer churn prediction model empowers the service providers with a suitable time interval to reform and apply a series of effective retention measures before the existing consumers migrate to other service providers. Customer relationship management (CRM) helps to synchronize, supervise, and manage the customers' interactions to maintain the customer satisfaction (*Retana, Forman & Wu, 2016*). On the other hand, customer segmentation plays a crucial role in order to perform effective customer analytics, which categorize customers into various groups according to their behaviors (*Bayer, 2010*). Customer's behavior analysis helps the operators to understand the reasons behind the customer's churn. It also helps the marketers to better understand their customers and update their policies according to the personalized requirements of service holders.

Rather than predicting only the customer churn, the telecom operators require more detailed analysis about the impacting factors and reasons behind the customer churn. Customer churn prediction and segmentation are supposed to be integrated as bi-level optimization. This research work presents a customer analytics framework for managing the customers churn (considering telecom sector as a case study) and aims to deploy efficient resource management for improved customer retention plans. A mixed data over-sampling method is presented to handle imbalanced class distribution problem of

customer churn datasets. Bayesian analysis was carried out in order to integrate the customer churn prediction module with that of customer segmentation module. The factors of each cluster were analyzed to provide recommendations to the operators and decision makers and to set the foundation for further measures in the future. In order to carry out the research study and experimentations, the following assumptions were made:

- Data availability assumption: It is assumed that sufficient and relevant data related to IoT device usage, service packages, and customer behavior are available for analysis.
- AI and IoT integration assumption: The research assumes the successful integration of artificial intelligence (AI) and Internet of Things (IoT) technologies within the telecom sector, as these technologies are central to the study.
- Churn and customer segmentation relevance assumption: The study assumes that churn recognition and customer segmentation are critical factors for customer retention in the telecom sector.
- Bi-level optimization feasibility assumption: The proposed unified customer analytics platform assumes that it is feasible and beneficial to address churn recognition and customer segmentation as a bi-level optimization problem, improving overall accuracy.

## RESEARCH MOTIVATION AND CONTRIBUTION

Nowadays, telco users have many options as the services and products are updated constantly. Therefore, it is pivotal for the operators to maintain the existing users with their requirements because the process of gaining new customers is five to 10 times costly than maintaining the existing ones. Also, the high service assumptions of customers generate the high marketing costs (*Olle, 2014*). Therefore, the use of AutoML methods for efficiently analyzing the customers' data and maintaining the retention strategies of existing customers is the hot spot in the telecommunication industry.

Most studies consider churn recognition and segmentation as two independent tasks which reduce the overall accuracy of the system. An isolated churn prediction module cannot find actual reasons behind the customer churn, since the service providers can only interpret which consumers are most likely to turnover. On the other hand, if customer segmentation is carried out individually, the service providers can only find the customer characteristics of various clusters and it is not possible to know which customers are most likely to churn. For effective marketing strategies, it is necessary to find out the actual reasons behind customer churn (*Khalatyan, 2010*; *Agrawal et al., 2018*). Another crucial problem for churn prediction is that the majority of customers in telecommunication want to stay with telco operators for a long period of time and keep up to enjoy the services rather than to transfer to another service provider. This behavior causes the customers churn data to be highly imbalanced.

The aforementioned drawbacks motivated us to integrate the customer segmentation and customer churn prediction as a single full fledge solution for building more efficient retention strategies. Bayesian analysis is applied in order to conduct the factor analysis and used as a connector between customer churn prediction and segmentation modules. As an

intermediary process, the key factors contributing to the customer churn are fed to segmentation module to group the customers more efficiently. With the help of segmenting the customers in several groups, our method makes personalized decision making to facilitate each individual group of users to ensure their stay with the company. Personalized decision-making plays a crucial role and involves tailoring recommendations and strategies for individual customers based on their unique behavior and characteristics. Here is a discussion of personalized decision-making:

*Customer-centric approach*: The emphasis on personalized decision-making underscores a customer-centric approach. Instead of applying uniform strategies to all customers, our method recognizes that different customers have distinct preferences, needs, and behaviors.

*Behavioral segmentation*: To enable personalized decision-making, the research first segments customers behaviorally. This segmentation likely involves categorizing customers into different groups based on their historical interactions, usage patterns, and responses to services.

*Recommendation system*: Once customers are segmented, personalized recommendations are generated. These recommendations are tailored to each customer group, taking into account their specific preferences and behaviors. For instance, a group of customers who frequently use certain IoT services might receive recommendations related to those services.

*Enhancing customer retention*: Personalized decision-making is instrumental in enhancing customer retention. By providing customers with services and offers that align with their preferences, the likelihood of retaining them increases. Our approach aims to minimize churn by addressing individual customer needs effectively.

*Data-driven insights*: Achieving effective personalized decision-making relies on analyzing extensive customer data. The study likely employs data-driven insights to understand customer behavior, preferences, and response to different strategies.

*Decision-maker benefits*: The personalized recommendations are not only advantageous for customers but also benefit decision-makers within the telecom sector. These recommendations provide valuable insights into how to optimize service packages and offerings for various customer segments.

The rest of the article is organized as follow. In the introduction section, we provide insights into the research motivation and outline the contributions of this study. "Research Motivation and Contribution" offers an in-depth exploration of the relevant literature, providing valuable context for our work. "Literature Review" delves into the methodologies and materials employed in our research. Moving forward to "Material and Methods", we rigorously validate, analyze, and discuss the applied methods, presenting our findings and insights. Finally, in "Results and Discussion", we draw conclusions based on our study's outcomes and offer suggestions for future research directions.

## LITERATURE REVIEW

### Churn prediction

In most of the recent studies, the use of predictive machine learning and deep learning models with multiple pre-processing techniques has been explored for solving customer

churn prediction problems. A Multilayered Neural Network (MNN) was designed (*Agrawal et al., 2018*) for the demonstration of the customer churn prediction on the IBM Watson Telco Customer Churn Dataset (data with 7,043 customers). The method was built upon the assumption that the categories with higher value are better and achieved the accuracy of about 80%. Extreme gradient boosting (XGBoost) based on the churn prediction algorithm (*Khanna et al., 2020*) was proposed for the Telecom Dataset (with 21 features and 3,333 samples) and turned out to perform better among other existing models. The imbalance data problem is handled with an oversampling method called SMOTE and the correlation between variables is used for feature selection. The model emphasized on accurately predicting the churners as compared to non-churners and achieved true positive (TP) rate of 81%.

*Li & Zhou (2020)* studied several classification approaches in order to predict the customer churn. The data used in their study was derived from database (including 437 fields) of customer basic and behavioral information. Support vector machine (SVM), Random-Forest (RF), linear regression (Lin.R) and logistic regression (LR) are used to test the current and historical data of customer information. Targeted variable is transformed into normal distribution to enhance the accuracy. Two evaluation metrics were used; namely, accuracy and coverage. For these two measures, the achieved results of the model are 82% and 42%, respectively. It has been observed that formulation of hierarchical customer retention strategies can reduce the cost of customer packages and enhance the efficiency of customer relationship management.

Oversampling and under-sampling techniques (*Ahmad, Jafar & Aljoumaa, 2019*) are explored for solving the imbalance data problems in telco industry. Most of the works used SMOTE as an oversampling technique for handling the imbalance data problem, but SMOTE can only deal with continuous data features (*Nguyen & Duong, 2021*). To overcome this limitation, SMOTE-ENC (*Mukherjee & Khushi, 2021*) and SMOTE-NC (*Rahmayanti, Saifudin & Ana, 2021*) are proposed as a variant of SMOTE to deal with both continuous and categorical features. Various over-sampling methods were analyzed by *Amin et al. (2016)* in order to handle the class imbalance problem for customer churn prediction. These six methods are: Adaptive Synthetic Sampling Approach (ADASYN), Couples Top-N Reverse k-Nearest Neighbor (TRk-NN), Immune Centroids Oversampling Technique (ICOTE), Mega-trend Diffusion Function (MTDF), SMOTE, and Weighted Minority Oversampling Technique (MWMOTE). They examined these methods on four publicly available datasets to predict customer churn, along with four rules-generation algorithms (LEM2, Covering, Exhaustive, and Genetic algorithms). Experimental analysis indicated that MTDF with genetic algorithm achieved best results. Under-sampling methods are also used to handle imbalance data problem. Single under-sampling method can cause the biasness when choosing the majority class. *Salunkhe & Mali (2018)* proposed a hybrid approach based on SOS-BUS that makes the combination of both the BUS and SMOTE techniques. In few experimental setups, the results were enhanced using only the SMOTE on imbalance dataset rather than SOS-BUS combination. *Tang et al. (2020)* proposed a hybrid approach based on XGBoost and Multilayer perceptron (MLP) to predict the customer churn. The authors used two datasets: (1) a music dataset (with 14

features), and (2) a telecom dataset (with 57 features). The process was divided into two stages. In the first stage, XG Boots was used to extract the leaf number of customers from numerical features. In the second stage, MLP was used to deal with leaf and discrete features. Comparative analysis reveals that the model achieved accuracy of about 96% and 72% on music and telecom datasets, respectively. The method achieved the F1-score of about 62% and 20% on these two datasets, respectively. *Wang, Nguyen & Nguyen (2020)* proposed machine learning models with ensemble learning approaches. The study utilized the Kaggle dataset with 99 features. The individual models including DT, RF, SVM, Naïve bayes (NB), MLP, k-NN and soft-voting based ensemble models were used for comparative analysis of the study. Results demonstrate that XGBoost and soft-voting based ensemble models performed well on customer churn prediction problem with 0.68% of AUC score.

In *Pamina et al. (2019)* classification approaches were applied to identify the factors that are highly influencing on customer churn. The IBM Watson dataset was used in the study for the churn prediction using multiple machine learning approaches including k-NN, RF and XGBoost. Correlation values were used for the selection of best features. Comparative analysis highlighted that XGBoost performs well on the IBM dataset in terms of accuracy and F1-score of 0.79% and 0.58%, respectively. For the exploration purpose of different data conversion methods and training models for cross-company churn prediction (CCP), the study (*Amin et al., 2019*) conducted the comparison in CCP domain. The classification methods used for experiments include NB, k-NN, GBT, and Deep Neural Network DNN. The results demonstrate that NB achieved high AUC value of 0.51.

*Panjasuchat & Limpiyakorn (2020)* proposed a Reinforcement Learning based model to predict the customer churn. A Deep Q Network (DQN) was applied on acquired dataset for the classification purpose. The dispensation of dataset is simulated differently including shuffle sampling to accept and adapt to the changes in method. The study utilized the telecom customer dataset from Kaggle with 99 features. Comparative analysis with traditional machine learning models demonstrated that proposed DQN achieved an accuracy of 65.26% and 55.04% on original and shuffled dataset, respectively. DQN also performed well in terms of other evaluation measures including F1-score, precision and recall on original and shuffled datasets.

*Ahmed & Maheswari (2019)* proposed an enhanced ensemble classifier model with cost heuristic based uplift modeling. Based on various algorithms, a diversified ensemble model was constructed in order to achieve the first order predictions of telco customer. Later, the predictions were further passed to a heuristic-based combiner for further processing to achieve better forecasting results. Comparative analysis highlighted that the proposed ensemble model is 50% more cost effective than existing ensemble models.

## Customer segmentation

Researchers used various unsupervised learning techniques for customer segmentation based on behavioral and factor analysis. *Namvar, Ghazanfari & Naderpour (2017)* proposed the data-driven segmentation for obtaining the increment in Average Revenue Per User (ARPU), in order to help the operators to design their marketing strategies.

K-mean clustering algorithm was used to divide the process into two segments (1) behavioral segmentation, and (2) beneficial segmentation. K-means clustering is careful regarding high variance of data; however, user need to prespecify the value of 'k'.

Decision tree (DT) with naïve Bayesian modeling was used (*Dullaghan & Rozaki, 2017*) for customer profiling segmentation based on customer demographics, sales information and billing behaviors. However, naïve Bayes assumes the independent predictor features. Limited factors were considered for analysis, so the operators are not able to understand the customers' behaviors and reasons behind the churn. In *Bayer (2010)*, the authors built an intuition that using value segmentation along with behavioral segmentation, the customer base can be divided into various levels of values and behavioral clusters. Each of these segments were further characterized using their features *e.g.*, likelihood, stickiness, promotion score, and average score.

An innovative customer segmentations approach based on customers life cycle was developed in *Han, Lu & Leung (2012)* in order to recognize some high valued customers. These values were divided into long term values, direct values, indirect values, current values, and historical values. The computation of these values composed five models. These five models were allocated weights by experts and high valued customers were pointed out through ranking.

In a study (*Faris, Al-Shboul & Ghatasheh, 2014*), an effective framework was proposed to boost up the capability of predicting the customer churn in order to maintain the customer retention strategies. The framework was based on the combination of two heuristic approaches: (1) Self organizing map (SOM), and (2) genetic programming (GP). The study utilized the real time customer dataset that was provided by a local telecom operator. Initially, SOM was applied on acquired dataset in order to analyze the clusters of various types of customers and then the outliers were removed from clustered data. Finally, GP was applied for constructing the effective path for the prediction of customer's churn.

## Integrated churn prediction and customer segmentation

In a recent baseline approach, *Wu et al. (2021)* used an oversampling method called SMOTE to the customer churn data. The method cannot handle mixed data which is its major limitation. Important features were selected using the chi-square test. The classification models including LR, DT, NB, AdaBoost, RF and MLP were used for prediction. The K-mean algorithm was used for customer segmentation. However, k-mean algorithm requires the predefined value of 'k' and it cannot detect the noisy points of data samples. The F1-score was dropped as compared to accuracy after applying SMOTE due to its limitation of handling mixed types of attributes.

Churn prediction model was proposed in *Ullah et al. (2019)* that used classification and clustering methods to identify the customer's churn. The authors used two datasets: (1) the GSM telecom service provider (with 29 features and 64,107 samples), and (2) the churn-bigml dataset (with 16 features and 3,333 samples). Information gain and correlation score were used for the selection of features. Results demonstrated that RF performs well as

**Table 1 Comparative analysis of recent methods for churn prediction and customer segmentation.**

| Author(s) | Methodology | Features | Challenges |
|---|---|---|---|
| *Lewaaelhamd & Israa (2023)* | K-means and DBSCAN clustering | Applies machine learning and RFM analysis for churn prediction using transactional data and evaluates k-means and DBSCAN clustering for customer segmentation into six clusters. | Due to limited data availability, often relying mainly on transactional data, accurate customer churn analysis may be impeded by the absence of comprehensive consumer data. |
| *Tran, Le & Nguyen (2023)* | LR, DT, SVM | Utilizes a variety of machine learning models, including k-means clustering for customer segmentation, k-nearest neighbors, logistic regression, decision tree, random forest, and support vector machine for customer churn prediction. | Lack in exploring techniques to handle imbalanced data in churn prediction. |
| *Demir, Buse & Övgü Öztürk Ergün (2023)* | LR, SVM, RF, XGBoost. | Focuses on mitigating customer churn in the banking sector, analyzing data to predict potential defectors and assessing model effectiveness using various metrics. | Restricted access to large banking datasets hampers generalizability. Current attributes don't capture behavior right before churn, limiting accuracy. Manual process in the app is time-consuming and less efficient. |
| *Prabadevi, Shalini & Kavitha (2023)* | Stochastic gradient booster, random forest, LR, k-NN | Focuses on client retention as a critical factor affecting a company's revenue. Aims to identify and predict client churn early to enable proactive measures to retain customers. | Need focus on refined data-side preprocessing, exhaustive hyperparameter standardization, and advanced optimization methods for hyperparameter tuning. |
| *Asfaw (2023)* | RF, LR, gradient boosting, XGBoost, and light gradient boosting | The paper focuses on predicting customer churn in the banking sector, recognizing that customer churn and engagement are crucial concerns for banks in a competitive market. | Imbalance class problem is highly affecting the performance need more model experiments to deal class imbalance problem |
| *Zelenkov & Suchkova (2023)* | Recency frequency, and monetary (RFM) | The paper underscores the significance of customer retention for businesses and the need to allocate retention resources based on the potential profitability of customers. Proposes an extension of the RFM model by incorporating estimates of the probability of changes in customer behavior, aiming to enhance churn prediction accuracy. | The use of extended sample duration (over 2 years) could hinder the accurate determination of average coefficients and trend analysis. Exploring more complex models than logistic regression might introduce unnecessary complexity for the paper's goals. |
| *Panimalar, Arockia & Krishnakumar (2023)* | DFE-WUNB (Deep Feature Extraction with Weight Updated Tuned Naïve Bayes) | Acknowledges the increasing importance of minimizing customer churn in a competitive market, highlighting the need for an effective Customer Churn Prediction (CCP) model. Proposes the use of a DFE-WUNB (Deep Feature Extraction with Weight Updated Tuned Naïve Bayes classifier) churn prediction model in a cloud-computing environment. | Complex and computationally intensive. Sensitive to the hyperparameters of the ANN and the Naïve Bayes classifier. |
| *Shobana et al. (2023)* | SVM | This research emphasizes ecommerce customer retention amid growing competition and customer acquisition costs. It stresses the need for understanding churn reasons and implementing personalized win-back strategies using various customer data. | Explore only SVM model. Use of other complex machine learning models can lead the performance better. |

compared to other classification algorithms in terms of 88.63% of rightly classified instances. In Table 1, we have compared some of the recent methods related to churn prediction and customer segmentation and summarized their salient features along with major limitations.

Our method is built upon the limitations of the existing methods and compete against the baseline approach (_Wu et al., 2021_) keeping in view various dimensions. The main contributions are summarized in follows:

- An oversampling method is presented which can handle mixed data and is also suitable for handling highly imbalanced-class distribution problem of telecom datasets.
- Factor analysis module is designed to govern the process of customer behavioral segmentation and where the user need not to prespecify the number of clusters?
- Customer churn prediction and segmentation are integrated as a single solution for achieving better insights into customer's analytical datasets.
- Personalized recommendations are generated for each group of customers to help decision makers in effective service management for the given customers.

## MATERIALS AND METHODS

This section provides the detail of selected datasets, preprocessing steps, features selection method, and design stages of proposed method.

### Datasets

In this research, three different datasets are utilized for experimental analysis. There are many online repositories which contains publicly available datasets for customer churn analysis from multiple open sources. The selected Dataset 1 from IBM is collected from https://www.kaggle.com/blastchar/telco-customer-churn. The publicly available Dataset 2 is extracted from the current Kaggle telecom churn prediction competition 2020 https://www.kaggle.com/c/customer-churn-prediction-2020/data. Dataset 3 (Cell2Cell dataset) is provided by Teradata center of CRM at Duke University and the link is available as https://www.kaggle.com/jpacse/datasets-for-churn-telecom. The summary of all these datasets is presented in Table 2.

### Proposed methodology

In this study, a unified customer analytical framework is proposed as illustrated in Fig. 1. The method consists of several interrelated phases. The detail of each phase is given in follows.

#### Data preprocessing

Data preprocessing is the first and essential part of the proposed framework. The aim of this step is to upgrade the quality of the collected data. Data cleaning is the basic step to refine data from noise, missing values, and irrelevant data. The exploration of the collected datasets revealed that these contain missing values and outliers. Basic machine learning models are not able to work with missing data and most of the models are not suitable for the outliers (_Ribeiro, Seruca & Durão, 2017_).

Figure 1 shows that the phase of data pre-processing consists of handling imbalanced data, data normalization and handling missing values. Data preprocessing is followed by EDA (exploratory data analysis) which is helpful to intelligently understand the data and

**Table 2 Dataset details.**

| Datasets | Dataset name | Total features | Total instances | Class imbalanced ratio |
|---|---|---|---|---|
| Dataset 1 | IBM | 21 | 7,044 | No: 5174, Yes: 1869 |
| Dataset 2 | Kaggle Telco 2020 | 20 | 4,251 | No: 3652, Yes: 598 |
| Dataset 3 | Cell2Cell | 58 | 51,048 | No: 36336, Yes: 14711 |

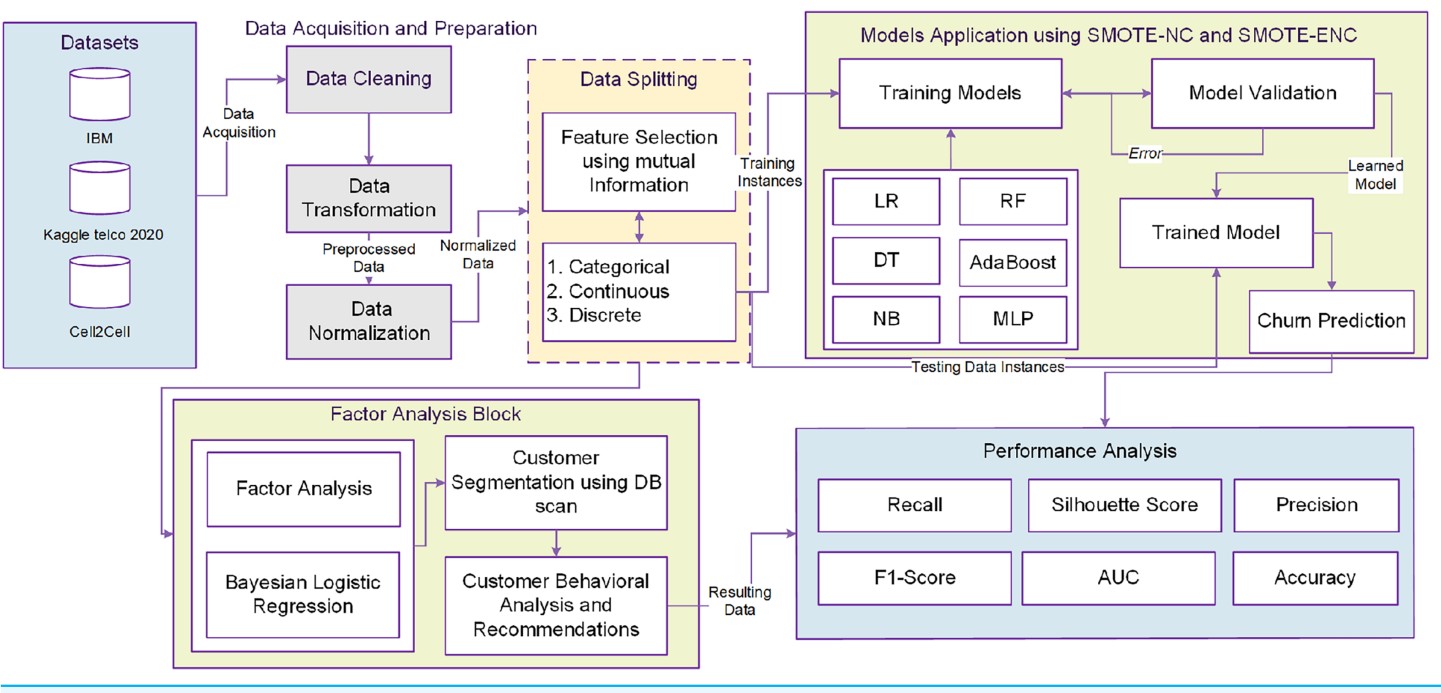

**Figure 1 Proposed framework.**               

each attribute before feeding them into ML models. Data normalization and transformation integrates and unifies the importance of all data features. In this study, scaling of all features is done using the min-max scaler technique. All values are shifted and rescaled in the range between 0 and 1. Mathematical representation of min-max scaler is illustrated in Eq. (1). Here, X is the original feature vector whereas X' is the scaled feature vector.

$$X' = \frac{X - min(X)}{max(X) - min(X)} \tag{1}$$

### Features selection

High dimensional data analysis and exploration is an extensive problem for researchers in machine learning domain. Useful features selection provides an effective way in order to reduce the irrelevant and insignificant data features. As shown in Fig. 1, after data preprocessing, mutual information (MI) is applied for feature selection of customers data.

The datasets contain the features of different types *e.g.*,: categorical, discrete, and continuous valued. Mutual information often uses the general form of correlation coefficient and measures the dependency between random variables. The process starts by adding features X from those containing most of the information of the target variable Y. MI between target variable Y and input variable X is calculated as indicated in Eq. (2).

$$I(X; Y) = H(Y) - H(X|Y) \tag{2}$$

Here H (Y) is the measure of target class information and H (X | Y) is the conditional entropy for X given Y. The I (X; Y) is the MI value between the set of features X and target class Y.

The dataset becomes imbalanced when the classification categories are not equally distributed or when the target class contains irregular distribution of observations. Achieving higher classification accuracy becomes difficult because of the unbounded and imbalanced nature of the classes. The datasets of telecom customer are highly imbalanced. When training over the data which contains the class imbalance problem, the ML models will overclassify the majority class. Mostly, the classifiers focus on majority class rather than misclassifying or ignoring the minority class (*Kaur, Pannu & Malhi, 2020*; *Sana et al., 2022*; *Saha et al., 2023*). Therefore, for acquiring better and accurate results, we need to handle the class imbalance problem.

### Data balancing with over-sampling methods

Most of the oversampling methods for churn prediction use SMOTE for handling class imbalance problem. However, SMOTE can only deal with continuous valued features (*Wu et al., 2021*). The telecom customer datasets contains both categorical and continuous variables. Auto Machine Learning (AutoML) oversampling techniques, including SMOTE-ENC (Synthetic Minority Oversampling Technique with Encoded Nominal and Continuous features), fall under the AutoML umbrella due to their autonomous and adaptable nature. These methods serve a crucial role in addressing class imbalance within datasets, particularly when dealing with datasets that feature both categorical and numerical attributes. What sets AutoML oversampling apart from traditional techniques is its ability to function with minimal human intervention. It automates various aspects of the oversampling process, such as encoding features to handle both categorical and numerical data effectively. Unlike manual methods that often require domain knowledge and expert judgment to make decisions about preprocessing steps, AutoML oversampling adapts to the specific characteristics of the input data. The automation in AutoML oversampling not only simplifies the preprocessing phase but also ensures consistency and efficiency in managing class imbalance across a wide range of datasets. This adaptability is particularly valuable when dealing with diverse and large-scale datasets in real-world applications. As shown in Fig. 1, modified versions of SMOTE-NC and SMOTE-ENC methods are applied on mixed type of features selected using mutual information for handling the class imbalance problem. The samples of minority class are generated synthetically (to balance its count with majority class) using median and nearest neighbors

of existing samples. For median calculation in SMOTE-NC, the median of standard deviations of all continuous features related with minority class is computed. This median is added into Euclidean distance due to the reason that the nominal features will contradict for a specimen and its nearest neighbors. Nearest neighbors are identified using the Euclidean distance between the feature vectors in which the k-nearest neighbors (k-NN) are being exposed from minority class samples and other feature vectors (minority instances) by adopting the continuous feature space.

In our method, the sampling strategy for SMOTE-NC is set to be float as this is suitable with binary classification. Float responds to the percentage of minority class instances over the number of majority class instances after resampling. Mathematically, it is represented as Eq. (3).

$$\alpha_{os} = N_{tm}/N_p \tag{3}$$

In Eq. (3), $N_{tm}$ demonstrates the number of minority class instances after resampling and $N_p$ shows the total number of majority class instances.

In SMOTE-NC, the inter-level distance of categorical features remains same as the distance of categorical features is always the median of standard deviation of continuous features. Comparatively, in SMOTE-ENC, the inter-level distance of categorical features is not always same because the distance of categorical features is not dependent on the continuous features of the dataset. In SMOTE-ENC, categorical features are encoded as numeric values and the diversity between these two numeric values reflect the quantity of change of cooperation with minority class. The mathematical representation of encoding of nominal features for minority class instances is illustrated in Eq. (4).

$$X = (o - \acute{e})/(\acute{e} * m) \tag{4}$$

where 'o' shows the number of minority class samples in training data and 'm' is the median of standard deviation of continuous features. The $\acute{e}$ can be calculated as given in Eq. (5).

$$\acute{e} = e * ir \tag{5}$$

in Eq. (5), 'e' shows the total number of minority class samples in training data and 'ir' represents the imbalance ratio of the training dataset. In this way, the presented SMOTE-ENC is able to encode the nominal features into numeric values for better interpretation of nominal features with specific considerations. After the encoding of categorical features, new synthetic data points are produced according to the majority of its k-nearest neighbors. For continuous features, SMOTE-NC and SMOTE-ENC generates new synthetic minority class samples in the same way as was done for SMOTE.

## Machine learning models and evaluation metrics
### Machine learning models
The presented oversampling methods of SMOTE-NC and SMOTE-ENC are applied to only the training part of the datasets in order to handle the class imbalance problem. Afterwards, six machine learning classifiers (Logistic Regression, Decision Tree, Random

**Table 3 Hyper-parameter configuration for all experiments.**

| Models | Hyper parameters | Dataset 1 | | Dataset 2 | | Dataset 3 | |
|---|---|---|---|---|---|---|---|
| | | SMOTE-NC | SMOTE-ENC | SMOTE-NC | SMOTE-ENC | SMOTE-NC | SMOTE-ENC |
| LR | Penalty | l2 | l2 | l2 | l2 | l1 | l1 |
| | Solver | Liblinear | Liblinear | Sag | Sag | Liblinear | Liblinear |
| DT | Max_feature | 15 | 15 | 15 | 15 | 10 | 10 |
| | Max_leaf_node | 20 | 25 | 45 | 20 | 35 | 35 |
| RF | N_estimator | 20 | 20 | 35 | 40 | 25 | 25 |
| | Max_feature | 15 | 15 | 15 | 15 | 20 | 20 |
| | Max_leaf_node | 25 | 25 | 45 | 65 | 45 | 45 |
| AdaBoost | N_estimator | 60 | 65 | 25 | 25 | 50 | 15 |
| | Learning rate | 1.0 | 1.0 | 0.6 | 0.6 | 1.0 | 1.0 |
| MLP | Solver | Adam | Adam | Lbfgs | Lbfgs | Adam | Adam |
| | Hidden_layer_size | (10,1) | (10,1) | (25,1) | (25,1) | (30,1) | (30,1) |

Forest, Naïve Bayes, AdaBoost, and Multi-layer Perceptron) are used over these synthetically balanced datasets. It is reported in the literature (*Li & Zhou, 2020*; *Wang, Nguyen & Nguyen, 2020*; *Amin et al., 2019*; *Demir, Buse & Övgü Öztürk Ergün, 2023*; *Prabadevi, Shalini & Kavitha, 2023*) that these classifiers are frequently applied and performed well in various studies related to churn prediction, so, these are chosen for a fair comparison. The hyper-parameters are tuned for all the experiments using grid search method and the selected values are summarized in Table 3.

For better customer segmentation, factor analysis is carried out using Bayesian logistic regression in order to figure out the major factors behind the customer churn. This procedure is known as Bayesian analysis, in which most influencing and useful features are extracted for effective and precise customer segmentation. As elaborated in Fig. 1, this study takes the advantage of Bayesian analysis as the in-between process to bridge the churn prediction and customer segmentation. As the main objective of this research is better churn management, the customer segmentation is held considering only churn class data. The density-based spatial clustering of applications with noise (DBSCAN) clustering algorithm is used for churn customer segmentation and properties of each cluster can then be acquired.

As factor analysis is helpful for accurate customer segmentation, we can analyze more specifically only the most relevant features related to customer churn. Initially, the Bayesian logistic regression uses the Bayes' theorem, in which prior distribution of all parameters is defined from normal distribution with mean and standard deviation. After that, different parameters are drawn from prior distributions and embedded into the logistic regression model to obtain the posterior distributions. A logistic regression model operates exactly as linear model and calculates the weighted sum of independent variables

by estimation of coefficients. Rather than continuous values, it returns the logistic function as given below:

$$log(x) = 1/(1 + e^{\wedge}(-x)) \tag{6}$$

At last, based on the results of churn prediction and customer segmentation, customer behavioral analysis is conducted. The variability of churn is visualized and the properties of individual churn segments are analyzed. It can be helpful for service providers and planners to appear along with various retention strategies on various segments.

### Evaluation metrics

For the assessment of model's performance, k-fold cross-validation is applied in which the original data is further grouped into various parts. The training part is utilized to train the model and the test part is utilized to assess the model's performance. Every individual test gives the corresponding results, and the overall result is the average result of all tests. For the proper evaluation of proposed methodology, accuracy, precision, recall, F1 measure, AUC and silhouette score are calculated and analyzed.

i. Accuracy

Accuracy is one of the most instinctive measures of execution. In the proposed research, it is the ratio of correctly classified churn and non-churn customers from the total customers.

$$Accuracy = (TP + TN)/(TP + FP + TN + FN) \tag{7}$$

ii. Precision

It is calculated in order to determine the ratio of truly positive values from the total number of positively predicted instances. For our problem, it is the measure of correctly classified churn customers that the model labeled as churn. Mathematically, it can be defined as follows.

$$Precision = TP/(TP + FP) \tag{8}$$

ii. Recall

Recall is required to find that how many positive instances are correctly classified. For our problem, it is the measure of how well our model has identified the total churn customers out of all samples of churn class. Mathematically, it can be defined as follows.

$$Recall = TP/(TP + FN) \tag{9}$$

iv. F1-measure

It is the harmonic mean of the precision and recall. Mathematically, it can be defined as follows.

$$F1 - Measure = (2.Precision.Recall)/(Precision + Recall) \tag{10}$$

v. AUC

It is simply the area under the curve which can be calculated using Simpson's rule. The highest the AUC score, the better the classifier is.

iv. Silhouette Score

It is used to measure the goodness of clustering techniques. Mathematically, it can be defined as follows.

$$Silhouette\ Score = (b - a)/max(a, b) \tag{11}$$

where 'a' is the average intra cluster distance and 'b' is the average inter cluster distance between all clusters.

# RESULTS AND DISCUSSION

The experimental results are presented for Dataset 1, Dataset 2 and Dataset 3 using mutual information with proposed SMOTE-NC & SMOTE-ENC techniques in comparison with baseline SMOTE (*Wu et al., 2021*) approach. MI is applied on all the datasets for the selection of representative features.

## Results of Dataset 1

Dataset 1 originally contains 21 features and the 16 selected features using MI are summarized comma separately as follows: (gender, partners, phone_services, tenure, Internet_service, online_security, online_backup, device_protection, technical_support, streaming_tv, streaming_movie, contracts, multiple_line, payment_method, monthly_charges, total_charges).

In Fig. 2, results of all three methods on the IBM dataset are reported considering various evaluation measures. From these results, it can be observed that SMOTE-NC improves the result on the IBM dataset in terms of all evaluation measures. Random Forest outperformed on the IBM dataset and achieved an accuracy, precision, F1-score and AUC of 79.24%, 57.82%, 65.25% and 85.79% respectively, while NB achieves high recall value of 83.24%. Selection of suitable evaluation measure in the confusion matrix is the major concern for the assessment of the models and FN (false negative) should be rewarded more concentration when the dataset is imbalanced. In the proposed study, FN indicates that how many of the churn customers are not successfully classified. The high value of precision indicates that many customers are correctly identified as churn customers and only a few samples are labeled as churn, incorrectly. The significantly improved value of recall shows that many customers are classified as churn customers from the total customers' base that actually churn. The higher value of accuracy and F1-score shows that the model is now more feasible as compared with baseline approach due to handling of mixed data.

Moreover, as compared to SMOTE-ENC, SMOTE-NC outperformed on the IBM dataset in terms of accuracy, precision, and F1-score. However, this does not mean that after using SMOTE-ENC the model is inadequate or worse. As compared to baseline SMOTE, the SMOTE-ENC based trained model is still more practical as it achieves the highest recall value of 92.60%. It is noticeable that the recall value is efficiently improved
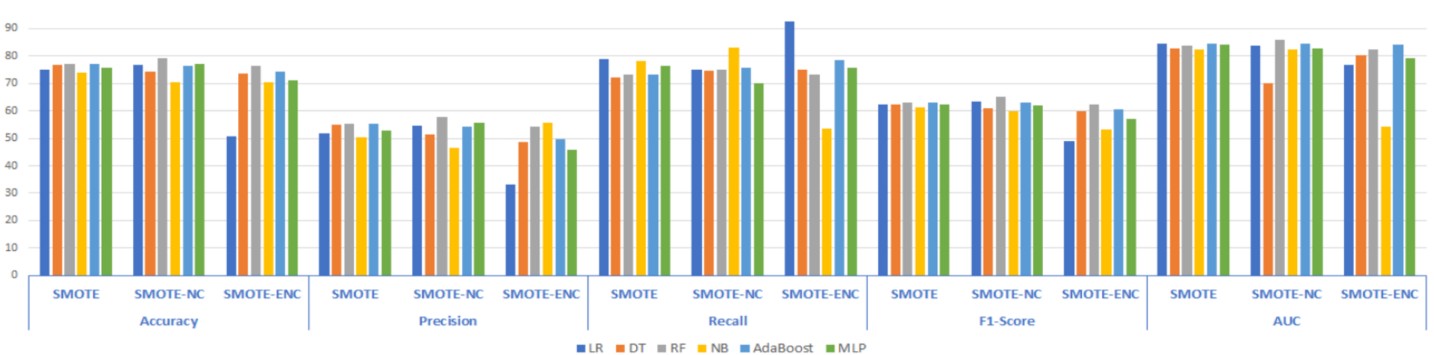

**Figure 2 Results with all the methods on the IBM dataset based on different evaluation measures.**

while applying SMOTE-ENC on the IBM dataset. Recall is comparatively important measure used for the assessment of the model which shows that the model is now able to address more customers who really turnover. As, SMOTE-ENC calculates the association between categorical attributes and target minority class, the SMOTE-ENC can perform significantly better as compared to SMOTE-NC, given that the dataset contains considerable number of categorical attributes and the association between these categorical attributes and target minority class is strong. In the IBM dataset, the affinity towards the categorical attributes and target minority class is analyzed as weak; this is why SMOTE-ENC is not able to perform well as compared to SMOTE-NC in terms of accuracy and F1-score. Considering the best model results of all the methods on the IBM dataset, it can be observed that as compared to SMOTE and SMOTE-ENC, SMOTE-NC significantly improves the results on the IBM dataset.

## Results of Dataset 2

Dataset 2 originally contains 20 features whereas 15 features are selected using MI is summarized comma separately as follows: (intl_plans, vmail_plans, number_vmail_messages, total_day_minutes, total_day_charges, total_eve_minutes, total_eve_calls, total_night_minutes).

In Fig. 3, results of all three methods on the Kaggle Telco dataset are reported considering various evaluation measures. These results demonstrate that SMOTE-NC efficiently handles the class imbalance problem of the Kaggle Telco dataset and achieves significantly better results as compared to SMOTE. Random Forest outperformed on the Kaggle Telco dataset and achieved high accuracy, precision, F1-score and AUC of 94.03%, 78.48%, 78.98% and 92.45% respectively. Logistic regression resulted in highest recall of 83.33%. From Fig. 3, it is depicted that as compared to SMOTE and SMOTE-NC, SMOTE-ENC highly improves the results on the Kaggle Telco dataset in terms of accuracy, precision, F1-score and AUC score. Random Forest outperforms other classification models in terms of all evaluation measures. RF achieved an accuracy, precision, recall, F1-score and AUC of 94.54%, 82.27%, 81.49%, 81.87% and 92.54%, respectively.

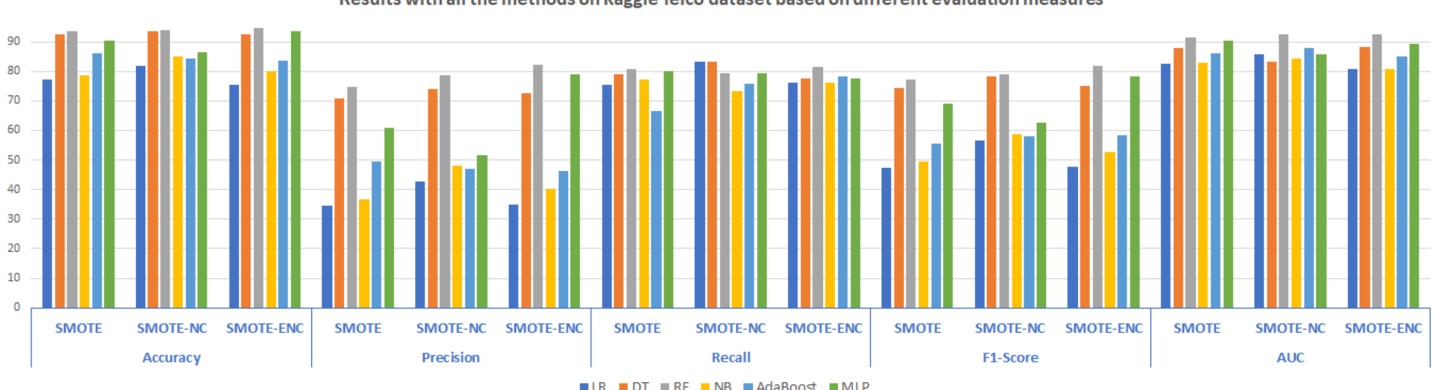

**Figure 3 Results with all the methods on the Kaggle Telco dataset based on different evaluation measures.**

SMOTE-ENC significantly improves the results on the Kaggle dataset and successfully identified more service consumers who are likely to quit, which is the major contribution of the proposed study.

## Results of Dataset 3

Dataset 3 originally contains 58 features. The 20 features selected using MI are summarized comma separately as follows: (made_calls_to_retention _teams, retention_call, current_equipment_day, monthly_revenue, retention_offer_accepted, credit_ratings, handsets_webcapable, active_subs, buys_via_mail_order, handset_model, off_peak_call_in_out, ageHH1, received_calls, homeownership, handset, peak_calls_in_out, ageHH2, total_recurring_charges).

In Fig. 4, the results of all the three models are summarized. The results demonstrate that SMOTE-NC significantly improves the results on Cell2Cell dataset in terms of the accuracy, precision, recall, F1 and AUC score. The results analysis of SMOTE-NC on the Cell2Cell dataset show that RF achieve significantly high accuracy, precision and AUC score of 69.57%, 44.54% and 63.28%, respectively. NB standing with high Recall and F1-score of 67.89% and 42.93% respectively. Improved values of precision and recall illustrates that as compared to baseline study, proposed model is able to identify more customers who are most likely to churn. Results analysis shows that, the model achieves high accuracy as compared to baseline approach which demonstrates that after applying SMOTE-NC on the Cell2Cell dataset, the overall performance of the classifier is significantly increased. The results show that as compared to SMOTE, SMOTE-ENC outperformed on the Cell2Cell dataset. After applying SMOTE-ENC on Cell2Cell dataset, RF achieves high accuracy, precision and AUC of 66.12%, 38.83% and 62.88% respectively. NB is standing with high recall of 75.02% and AdaBoost achieves a significantly high F1-score of 45.62%.

In SMOTE-NC, the inter-level distance of nominal attributes is consistently dependent upon the continuous attributes of the dataset. In particular, the distance of the categorical

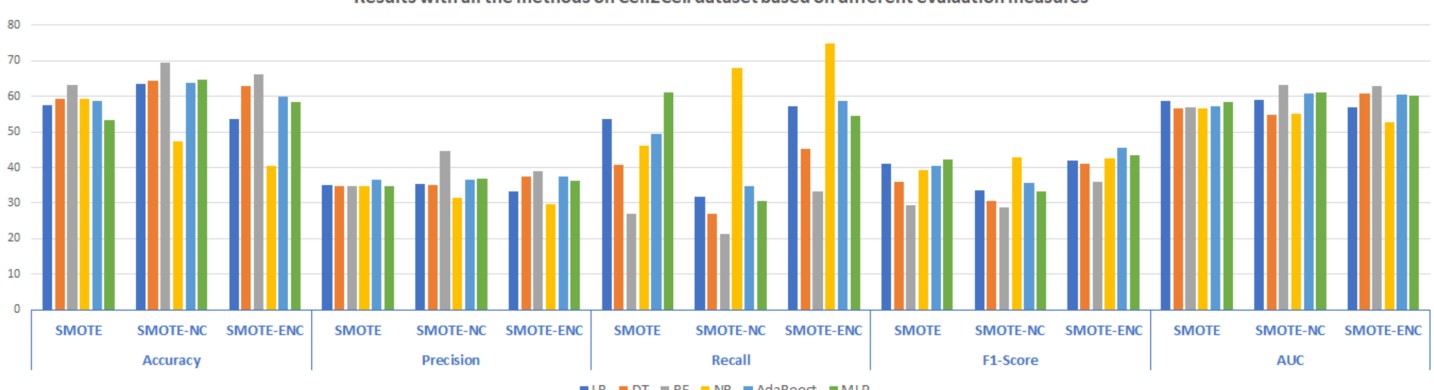

**Figure 4 Results with all the methods on the Cell2Cell dataset based on different evaluation measures.**

features is the median of the standard deviations of continuous attributes. Every label is analyzed to have the same tendency towards the target attribute, and hence mix-up the attribute's contribution regarding the distance calculation. In comparison, in SMOTE-ENC, the inter-level distance of categorical features is not dependent upon the continuous features of the dataset and capturing the categorical attributes contribution in distance calculation is done more accurately. Therefore, as compared to SMOTE and SMOTE-NC, SMOTE-ENC significantly improves the F1-score on the Cell2Cell dataset and identify more customers who are most probably to churn, which is very helpful for telecom service providers in order to retain the churn customers, efficiently.

## Average results analysis on all datasets

Average results analysis on all the considered datasets with all three methods are summarized in Fig. 5. It is illustrated that as compared to SMOTE, SMOTE-NC achieved high average results in terms of all evaluation measures. Random Forest outperformed in terms of accuracy, precision, F1-score and AUC score of 80.94%, 60.28%, 58.66% and 80.50% respectively. LR achieved high recall score about 63.41%. Analyzing the results, it can be observed that as compared to SMOTE and SMOTE-NC, SMOTE-ENC achieved significantly high average results in terms of recall and F1-score. RF with SMOTE-ENC outperformed in terms of accuracy, precision, F1 and AUC score of 80.04%, 58.46%, 60.20% and 80.22% respectively. LR standing with high recall score of 75.42%.

## Bayesian analysis and customer segmentation using DBSCAN

Factor analysis is conducted before the clustering in order to explore the influence of every component on customer churn by applying the Bayesian logistic regression model. After that customer segmentation is held for additional analysis of customers behaviors that gives various categories of customers properties for telecom service providers to manage the retention strategies.
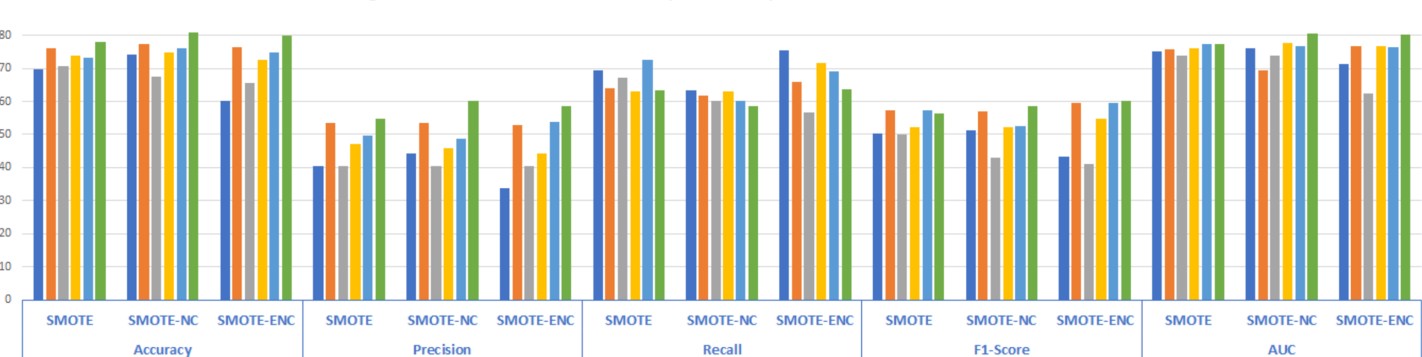

**Figure 5 Average results with all three methods on the (all datasets) based on different evaluation measures.**

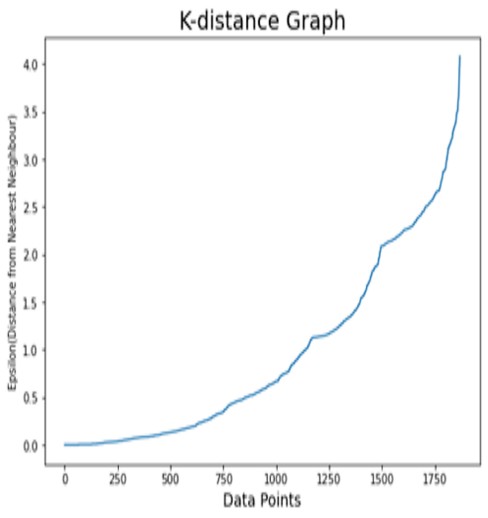

**Figure 6 K-distance graph of the IBM dataset.**

### Results analysis on dataset 1

K-distance graph is used in order to find the optimal value of epsilon (odd ratio as discussed below). For the IBM dataset, the appropriate value of epsilon is initiated to 2.8 as illustrated in Fig. 6. As logistic regression calculates the weighted sum of every considered feature, we use these weights of every individual feature in order to calculate the OR (odd ratio) to quantify the effect of every individual feature. The odds ratio is applied to calculate the correlative odds of the existence of the individual attribute. It is important to decide whether a specific factor is a risky factor or not for a specific target class. The effect percentage is utilized to measure different risky and protective factors for the particular target class. The positive range effect demonstrates that the specific factor is highly

**Table 4 Odds ratio and effect percentage of the IBM dataset.**

| Attributes | Weights | Odd ratio | Effects% |
|---|---|---|---|
| Gender | −0.1438 | 0.8660 | −13.4 |
| Senior_citizens | −0.0270 | 0.9734 | **−2.66** |
| Partners | 0.2429 | 1.2749 | 27.49 |
| Dependent | 0.0142 | 1.0143 | **1.43** |
| Phone_services | −0.1633 | 0.8493 | −15.07 |
| Multiple_lines | −1.0307 | 0.3568 | −64.32 |
| Internet_services | 0.1462 | 1.1574 | 15.74 |
| Online_security | 0.4711 | 1.6018 | 60.18 |
| Online_backups | −0.5439 | 0.5805 | −41.95 |
| Device_protection | −0.2866 | 0.7508 | −24.92 |
| Technical_supports | −0.1747 | 0.8397 | −16.03 |
| Stream_tv | −0.5268 | 0.5905 | −40.95 |
| Stream_movie | −0.0088 | 0.9912 | **−0.88** |
| Paperless_billings | 0.0021 | 1.0021 | **0.21** |
| Contracts | 0.3686 | 1.4457 | 44.57 |
| Payment method | −1.4117 | 0.2437 | −75.63 |
| Tenure | 0.1463 | 1.1575 | 15.75 |
| Monthly charges | −4.2216 | 0.0147 | −98.53 |
| Total charges | 2.3080 | 10.0543 | 905.43 |

Note:
  Bold values indicate less effective attributes to the churn.

**Table 5 Clusters sample size for the IBM dataset.**

| Dataset | Cluster 1 | Cluster 2 | Cluster 3 | Noisy points | Total |
|---|---|---|---|---|---|
| **IBM** | 1,435 | 127 | 103 | 204 | 1,869 |

correlated with target class (churn). The negative effect means that there are less chances of customer's churn. Odd ratio and percentage effects are calculated using Eqs. (12) and (13).

$$OddsRatio = e^{\theta} \tag{12}$$
$$Effect(\%) = 100 \cdot (OddsRatio - 1) \tag{13}$$

In Eq. (12), $\theta$ is the weight of individual attribute given by Bayesian Logistic Regression model. The odds ratio and effect percentages are calculated for each attribute and given in Table 4. The result illustrates that senior citizen, dependents, streaming movies and paperless billing are appeared as less effected on churn so, these attributes are discarded when performing the customer segmentation. The discarded features for customer segmentation are highlighted in the table. These features are further divided into three clusters by DBSCAN. The sample size of each cluster is depicted in Table 5. It is shown that cluster 1 has 1,435 sample size which is the largest sample from overall population. Cluster 2 contains 127 samples and cluster 3 has 103 samples, respectively. From Dataset 1,

**Table 6  Summary of clusters for the IBM dataset.**

| Attributes | C1 | C2 | C3 |
| --- | --- | --- | --- |
| Partners | High | Medium | Low |
| Phone_service | High | Low | Medium |
| Multiple_lines | High | Medium | Low |
| Streaming_tv | Medium | Low | High |
| Contracts | High | Low | Medium |
| Payment method (Through Check) | Low | Medium | High |
| Payment_method (Automated) | High | Low | Medium |
| Tenure | High | Medium | Low |
| Monthly_charges | High | Medium | Low |
| Online Security & backup | Medium | Low | High |
| Internet_service (DSL) | Medium | Low | High |
| Internet_service (Fiber optic) | High | Low | Medium |
| Total_charges | High | Low | Medium |

204 points are detected as noisy points by DBSCAN algorithm while doing customer segmentation. Noisy points are not related with any cluster and treated as outliers. The structured summary of clusters for the IBM dataset is illustrated in Table 6, where high labeled for a cluster means that this cluster contains the extreme number of customers for the particular attribute among the three clusters. Likewise, low means that the cluster has not many consumers related to the particular attribute and medium means average number of customers related to the specific attribute. For the partner feature, high demonstrates that the high number of customers are initiated as partners between all clusters. In the services related attributes, high for a cluster illustrates that high number of customers have used the services, among three clusters. All components of clusters for the IBM dataset are examined in Table 6. For payment method, high means that more customers use this type of method for payment. For contract and tenure related features, high represents longer working duration. In charges related features, high demonstrates the highest amount.

In cluster 1, customers have huge requirements and demands for services. This cluster highly adopted the fiber optic for internet services. Also, this cluster highly uses the automatic payment method. As the customers of cluster 1 have the prolonged contractual duration, they have stayed with the telecom company for a long period of time. The customers pay high charges and highly participated to the company's profit but, still they have churned. This cluster needs more observations and consciousness as they have high devotion and trustworthy for telco operators. As the customers related to clusters1 highly prefers the fiber optic and this type of service is much costly as compared to other internet services so, the operators need to provide the manageable and inexpensive services for their highly fiber optic users. Another recommendation for telco operators is that they can provide the high internet speed through DSL services so that if the customers are unable to

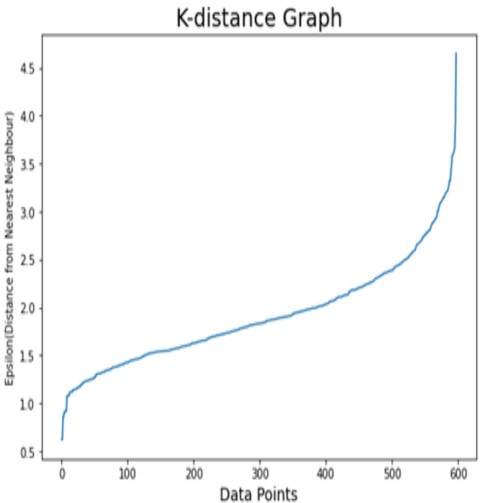

**Figure 7** **K-distance graph for the Kaggle Telco dataset.**

manage fiber optic charges, they can switch to another internet services in order to fulfill their needs rather than leaving the telecom company.

For internet services, customers of cluster 2 are not highly using the internet services and few customers are dependents. DSL is low budget and low speed internet service. The customers of cluster 2 have least contractual period and also hardly uses the automatic payment methods. Therefore, this type of customer does not have the ability to stay with any telecom operator or company for a long period of time as their contract duration and tenure both are not prolonged. This cluster is not much contributing to the company's revenue. This type of customers may be the new one's who just join the company for analyzing their services and tends to turnover for more cost-effective services. If the service providers establish the least cost services for these customers, then they will be able to build the trust of these customers for company's services and attract them to stay with the company for a long period of time. Cluster 3 has the high insistence for basic services as compared to cluster 2. For internet services, they mostly use the DSL but some of them prefer the fiber optic services as well. Few of them prefer the automatic payment method but most of them use the alternative method for online payments. As compared to other clusters, cluster 3 have less tenure and also don't have high contracts durations.

According to the services demands, cluster 3 is examined as the average cluster among cluster 1 and cluster 2. Their monthly and total charges are also not much high, so, this cluster may have not highly participated in company's revenue and not considered as the loyal customers for telco service providers. According to the 'charges' behavior of this cluster, the operators need to offer the reasonable economical packages for these customers, so that the customers can fulfill their needs in reasonable cost and contributes to the company's revenue. In this way, the operators will be able to maintain the retention strategies among these customers.

**Table 7 Odds ratios and effect percentage for the Kaggle Telco dataset.**

| Attributes | Weight | Odd ratio | Effects% |
|---|---|---|---|
| Account_length | 0.0320 | 1.0325 | **3.25** |
| Area_code | −0.0231 | 0.9772 | **−2.28** |
| Intl_plans | 2.0268 | 7.5898 | 658.98 |
| Vmail_plans | −2.1640 | 0.1149 | −88.51 |
| Number_vmail_message | 1.4983 | 4.4741 | 347.41 |
| Total_day_minutes | 2.1919 | 8.9522 | 795.22 |
| Total_day_calls | 0.0424 | 1.0433 | **4.33** |
| Total_day_charges | 2.1772 | 8.8216 | 782.16 |
| Total_eve_minutes | 0.9867 | 2.6824 | 168.25 |
| Total_eve_calls | −0.5633 | 0.5693 | −43.07 |
| Total_eve_charges | 0.7583 | 2.1346 | 113.46 |
| Total_night_minutes | 0.6560 | 1.9271 | 92.71 |
| Total_night_calls | −0.6526 | 0.5207 | −47.93 |
| Total_night_charges | 0.4367 | 1.5476 | 54.76 |
| Total_intl_minutes | 0.4520 | 1.5715 | 57.15 |
| Total_intl_calls | −0.9494 | 0.3870 | −61.30 |
| Total_intl_charges | 0.5814 | 1.7885 | 78.85 |
| Numbers_customers_service_ call | 2.3601 | 10.5920 | 959.20 |

**Note:**
  Bold values indicate less effective attributes to the churn.

**Table 8 Clusters sample size for the Kaggle Telco dataset.**

| Dataset | Cluster 1 | Cluster 2 | Cluster 3 | Noisy points | Total |
|---|---|---|---|---|---|
| **Kaggle Telco** | 102 | 317 | 156 | 23 | 598 |

### Results analysis on dataset 2

For dataset 2, K-distance graph is also used in order to find the optimal value of epsilon. Fig. 7 demonstrates the k-distance graph for the Kaggle Telco dataset. For the Kaggle Telco dataset, the appropriate value of epsilon is set to 3.3. Before the customer segmentation, factor analysis for this dataset is depicted in Table 7 with effect percentage and odds ratios of features. The features including Area_code, total_day_call and accounts_length have the minimum effect on churn target class. So, these attributes are discarded while performing the customer segmentation. The features including Total_day_charges, numbers_customers_service_calls, and total_day_minutes are considered as the risk factors to churn class. The selected attributes are further divided into three clusters using DBSCAN. The sample size of each cluster for the Kaggle Telco dataset is depicted in Table 8. It is illustrated that cluster 1 has 102 customers from 598 total customers. Cluster 2 contains 317 customers which is largest sample size from total samples. Clusters 3 contains 156 customers whereas 23 points are detected as noisy points. The summarized structure of clusters for the Kaggle Telco dataset is illustrated in Table 9. For attributes related to

**Table 9 Summary of clusters for the Kaggle Telco dataset.**

| Attributes | C1 | C2 | C3 |
|---|---|---|---|
| International_plan | Medium | Low | High |
| Voice_mail_plan | Low | Medium | High |
| Number_vmail_messages | Low | High | Medium |
| Total_day_minutes & charges | Medium | Low | High |
| Total_eve_minutes & charges | High | Low | Medium |
| Total_eve_calls | Medium | High | Low |
| Total_night_minutes & charges | High | Low | Medium |
| Total_night_calls | Medium | High | Low |
| Total_intl_minutes & charges | Medium | High | Low |
| Total_intl_calls | Low | High | Medium |
| Number_customer_service_calls | Medium | High | Low |

service, high demonstrates that more customers use these basic services among all clusters. For messages related attributes, high demonstrates high number of communication records. For total minutes and charges, high stands for longer period of time and where customers spend more amounts on calls.

Let us analyze the results summarized in Table 9. The customers in cluster 1 do not have many requirements for all types of services. They do not make the rapid calls to customers service providers. Few consumers have the international plans and also spend the limited amount on international call charges. It is also observed that, this cluster have only limited number of evening and night calls but, there is the high number of call minutes along with high amount of call charges. This observation demonstrates the main reason behind the churn of these customers as they avail the company's services with high charges. This type of customers requires the special evening and night packages in reasonable amount for specific time. To overcome the churn rate of this cluster, the telecom operators need to manage these customers by providing the sufficient least cost packages in accordance with the customer's calling durations.

As compared to cluster 1, cluster 2 has high requirements for services. The customers make the high national and international calls with highest international minutes and charges. Their phone calls property is appeared as constant and frequent. High number of calls to customer services are done by the customers of this cluster. This can be observed that this cluster is contributed to company's profit as they have the high international calling behaviors with high charges. On the other hand, they can call the customer care service, more rapidly, so, they might be churn frequently because these customers are not highly satisfied with telco operators and their costly services. The telecom service providers can manage these customers by providing the specific discount and concession according to their international calling behaviors. *e.g.*: if service consumers can be entertained by hourly packages, then they will be able to make extensive calls at fix rates. In this way the consumer wants to stay with operators for a long period of time.

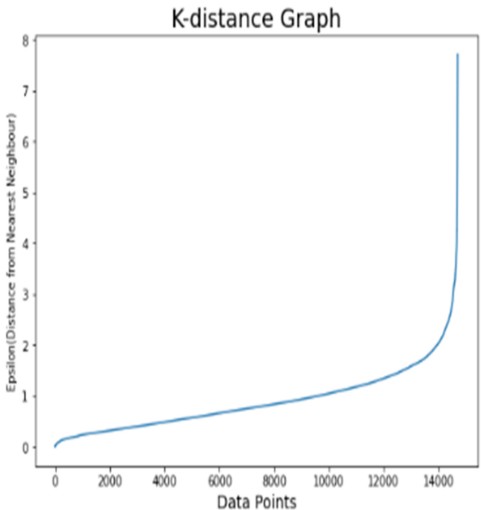

**Figure 8 K-distance graph for the Cell2Cell dataset.**

In cluster 3, mostly customers have not endorsed for international services and also their charges are not much high as compared to other depicted clusters. As compared to cluster 2, the customers of cluster 3 have high national call minutes and charges. But they hardly make the customer service calls that demonstrates that they are satisfied with service provider's feedback. This cluster seems to be more loyal with the telecom company and have stayed for a long period of time. As compared to cluster 1 and cluster 2, the customers of cluster 3 are not most likely to churn and also have the high contribution to company's profit. These are the most valuable customers for telecom operators.

### Results analysis on dataset 3

For the Cell2Cell dataset, K-distance graph (Fig. 8) is also used in order to find the optimal value of epsilon (and set to 1.8). Before the customer segmentation, factor analysis for dataset is depicted in Table 10 with odds ratios and effect percentage for all the features. It can be observed that buys_vmail_order, home_ownerships, peak_call_in_out and ageHH2 have minimum effect percentage on target class churn. Therefore, these attributes are discarded while performing the customer segmentation. The risk attributes of dataset 3 are current_equipment_days and retention_calls. Other listed attributes are considered as the protective factors.

After the factor analysis of the Cell2Cell dataset, segmentation is applied on the selected features. These selected features are further divided into three clusters using DBSCAN. The sample size of each cluster for dataset 3 is depicted in Table 11. It is illustrated that cluster 1 has 11,909 sample size which is the largest sample from overall population. Cluster 2 contains 1,711 samples and cluster 3 has 226 samples, respectively. From the Cell2Cell dataset, 864 points are detected as noisy points. The structured summary of clustering for the Cell2Cell dataset is illustrated in Table 12. The clustering analysis is concluded for the Cell2Cell dataset. In credit rating and calls related attributes, high demonstrates the high credit ratings, and longer call durations with high charges, respectively. For the handset_web_ capable and made_call_to_retention_team attributes, high demonstrates

**Table 10 Odds ratio and effect percentage for the Cell2Cell dataset.**

| Attributes | Weight | Odd ratio | Effects% |
|---|---|---|---|
| Retentions_call | 1.164 | 3.204 | 220.368 |
| Current_equipment_days | 1.369 | 3.931 | 293.063 |
| Monthly_revenue | 0.189 | 1.208 | 22.81 |
| Retentions_offer_accepted | −0.856 | 0.425 | −57.501 |
| Credit_ratings | −0.272 | 0.762 | −23.837 |
| Active_subs | −0.182 | 0.833 | −16.855 |
| Monthly_minutes | −0.451 | 0.637 | −36.301 |
| Total_recurring_charges | −1.372 | 0.254 | −74.650 |
| Received_calls | 0.458 | 1.581 | 58.107 |
| Made_calls_to_retention teams | 0.567 | 1.763 | 76.262 |
| Handsets_webcapable | −0.114 | 0.892 | −10.783 |
| Buys_vmail_orders | 0.039 | 1.040 | **3.987** |
| Handset_model | −1.276 | 0.279 | −72.079 |
| Off_peak_call_in_out | −0.317 | 0.728 | −27.160 |
| AgeHH1 | −0.475 | 0.622 | −37.812 |
| Home_ownership | −0.029 | 0.972 | **−2.820** |
| Handset | 0.824 | 2.280 | 128.006 |
| Peak_call_in_out | −0.052 | 0.949 | **−5.077** |
| AgeHH2 | 0.016 | 1.016 | **1.582** |
| Outbound_call | 0.452 | 1.571 | 57.114 |

**Note:**
Bold values indicate less effective attributes to the churn.

**Table 11 Clusters sample size for the Cell2Cell dataset.**

| Dataset | Cluster 1 | Cluster 2 | Cluster 3 | Noisy points | Total |
|---|---|---|---|---|---|
| Cell2Cell | 11909 | 1711 | 226 | 864 | 14711 |

the high number of customers is associated with the particular attribute. For active subs, high means that the more customers are subscribed to a particular service. All components of clusters for the Cell2Cell dataset are examined. The attribute AgeHH1 illustrates that cluster 1 contains older customers as compared to cluster 2 and cluster 3. They have associated with equipment days for extreme duration. They do not have longer call durations but they spend high amount on call charges. These customers are not highly web capable that demonstrates that they do not have the facility of internet service. They also have lower credit ratings as compared to other two groups. It is observed that this cluster rarely admired the retention offers. This means that the offers may not be highly attractive for them so that the customers might be switch to another company in order to find better services. The characteristics of this cluster illustrates that they are not the rapid mobile users, so, these customers need comparatively simple packages with lower amount. The telco operators need to propose such attractive retention offers according to the demand of

**Table 12 Summary of clusters for the Cell2Cell dataset.**

| Attribute | C1 | C2 | C3 |
|---|---|---|---|
| Retention_calls | Low | Medium | High |
| Current_equipment_days | High | Medium | Low |
| Monthly_revenue | Medium | Low | High |
| Retention_offers_accepted | Low | Medium | High |
| Credit_rating | Low | High | Medium |
| Active_subs | High | Low | Medium |
| Monthly_minutes | Low | Medium | High |
| Total_recurring_charge | High | Low | Medium |
| Received_calls | Medium | High | Low |
| Handset_refurbished | Low | Medium | High |
| Made_calls_to_retention _teams | Medium | Low | High |
| Handset_web_capable | Low | Medium | High |
| Off_peak_call_in_out | Low | Medium | High |
| AgeHH1 | High | Medium | Low |
| Outbound_calls | Low | Medium | High |

this cluster to overcome its churn rate. This cluster has participated in operator's revenue as charges are analyzed as high.

Cluster 2 contains the younger customers as compared to cluster 1 and have the medium level of call behavior. This cluster contains the high credit ratings. These customers avail the company's services with low charges. Also, their retention calls are not very high which means that they are satisfied with the retention offers and they want to stay for long duration with telecom operators. Their total monthly call minutes are high as compared to cluster 1, so, these customers have the high probability to contribute to company's revenue in contrast with cluster 1.

The customers of cluster 3 are more energetically attached with the retention campaign. This group contains more teenage customers than other two clusters. Along with middle level of credit rating, they also have the short period of time with current equipment days. This group requires the internet resources because extreme number of customers is using the web capable handset. This group has the characteristics of longest call durations and also, they make the calls more rapidly as compared to other groups. They have the highest monthly revenue and monthly call minutes, so, these customers are attributed as the most important customers that highly participated in company's revenue. They highly accepted the retention offers; however, they make the retention calls more often to customer service providers. This habit of cluster 3 demonstrates that the customers may be dissatisfied with some services, and this is the main reason behind their churn. The telecom operator needs to manage their basic services in order to fulfill the requirements of their customers and make some responding calls for the satisfaction of this group.

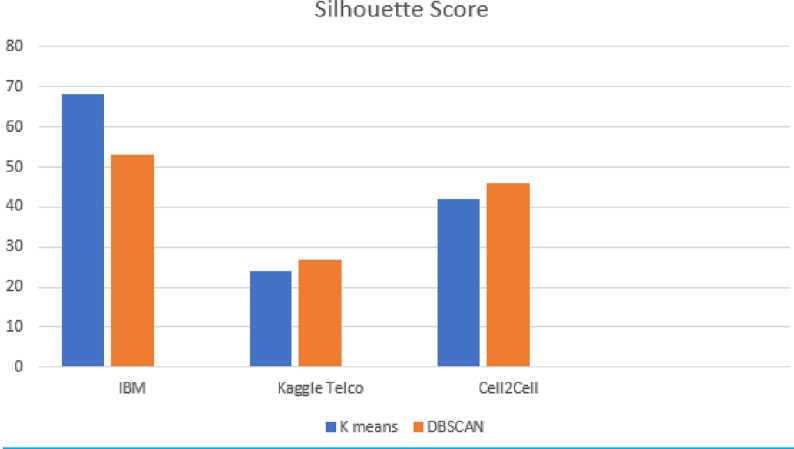

**Figure 9 Silhouette score analysis on all datasets.**

### Silhouette score analysis on all datasets

Silhouette score is applied for the evaluation of DBSCAN clusters. Figure 9 shows the results of silhouette score on all the datasets. From Fig. 9, it is observed that as compared to k-means, DBSCAN significantly improved the results on the Kaggle Telco and Cell2Cell datasets. K-means achieved a silhouette score of 24% and 42% on the Kaggle Telco and Cell2Cell datasets, respectively. After applying DBSCAN, silhouette score is increased by 26% and 45% on the Kaggle Telco and Cell2Cell datasets, respectively. Therefore, it is found that in comparison with k-means, DBSCAN is more efficient in forming the clusters for the Kaggle Telco and Cell2Cell datasets (as the mean intra cluster distance is minimum and mean inter cluster distance is maximum for the clusters of both the datasets). Further results analysis depicts that as compared to k-means, DBSCAN is not able to perform well on the IBM dataset because DBSCAN cannot handle the datasets consisting clusters of various densities. It was observed from the analysis that the IBM dataset contains the density variation between various data points so, DBSCAN is facing trouble with the IBM dataset when making the clusters.

## CONCLUSIONS

This research introduces an autonomous customer analytics framework addressing class imbalance in AIOT-based telecom churn datasets. Bayesian logistic regression was employed for customer segmentation, and personalized recommendations were tailored for each segment. The study eliminates insignificant features, providing precise churn customer segmentation and summarizing customer group characteristics. Notably, it extends existing research by integrating churn prediction and customer segmentation, offers a mixed data oversampling solution, and explores churn reasons for personalized recommendations. Experimental results demonstrate the framework's superiority over baseline methods. Random Forest with SMOTE-NC, outperforms all the competitors and achieves accuracy rates of 79.24%, 94.54%, and 69.57%, and F1-scores of 65.25%, 81.87%, and 45.62% for the IBM, Kaggle Telco and Cell2Cell datasets, respectively. In future,

hyper-parameters optimization using the Halving Grid search method will be studied to make the model more accurate. Moreover, random over sampling technique will be explored to handle the imbalanced dataset problem. Likewise, receiver operating curve (ROC) analysis will be further inspected in order to design the model for more accurate churn recognition.

### Funding

The work is supported and funded by the Princess Nourah bint Abdulrahman University Researchers Supporting Project number (PNURSP2024R407) and the Princess Nourah bint Abdulrahman University, Riyadh, Saudi Arabia. Dr. Salabat is working for National Research Foundation of Korea (NRF) under the Brain Pool Program (Grant No. 2022H1D3A2A02055024) and Creative Research Project (ID: RS-2023-00248526). The funders had no role in study design, data collection and analysis, decision to publish, or preparation of the manuscript.

### Grant Disclosures

The following grant information was disclosed by the authors:
Princess Nourah bint Abdulrahman University Researchers Supporting Project number: PNURSP2024R407.
Princess Nourah bint Abdulrahman University.
National Research Foundation of Korea (NRF): 2022H1D3A2A02055024.
Creative Research Project: RS-2023-00248526.

### Competing Interests

The authors declare that they have no competing interests.

### Author Contributions

- Ghulam Fatima conceived and designed the experiments, performed the experiments, performed the computation work, prepared figures and/or tables, and approved the final draft.
- Salabat Khan conceived and designed the experiments, prepared figures and/or tables, authored or reviewed drafts of the article, and approved the final draft.
- Farhan Aadil analyzed the data, authored or reviewed drafts of the article, and approved the final draft.
- Do Hyuen Kim analyzed the data, authored or reviewed drafts of the article, and approved the final draft.
- Ghada Atteia analyzed the data, prepared figures and/or tables, and approved the final draft.
- Maali Alabdulhafith analyzed the data, prepared figures and/or tables, authored or reviewed drafts of the article, and approved the final draft.

## Data Availability

The code is available in the Supplemental File.

The data is available at Kaggle:

- IBM. (2018). Telco Customer Churn. https://www.kaggle.com/blastchar/telco-customer-churn

- Kaggle. (2018). Customer Churn Prediction 2020. https://www.kaggle.com/c/customer-churn-prediction-2020/data

- Cell2Cell. (2018). Telecom Churn (Cell2Cell). https://www.kaggle.com/jpacse/datasets-for-churn-telecom.

## Supplemental Information

Supplemental information for this article can be found online at http://dx.doi.org/10.7717/peerj-cs.1756#supplemental-information.

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
