# Peer review of "An autonomous mixed data oversampling method for AIOT-based churn recognition and personalized recommendations using behavioral segmentation"

_PeerJ Computer Science, doi:10.7717/peerj-cs.1756_

## Round 0.1 · original submission · Major Revisions

The approach adopted in your research is novel and relevant, and hence deserves being published. However, the paper still needs some major improvements for for publication in PeerJ.

For example, the literature review must be significantly improved to include the more recent published works on churn prediction and customer segmentation.
And all the assumptions underlying the research work are not clearly explained and justified. The results ought to be better organised and presented so as to enhance clarity. The authors should use more graphs to to better compare results. 

Moreover, the manuscript should be proofread (also fixing the grammar mistakes) by an expert writer of English. In addition, I suggest using some tools such as Grammarly or DeepL.

Moreover, all the inquiries made by Reviewer 1 should be thoroughly addressed in the resubmitted version of the manuscript.

Reviewer 1 has suggested that you cite specific references. You are welcome to add it/them if you believe they are relevant. However, you are not required to include these citations, and if you do not include them, this will not influence my decision.

**Language Note:** The Academic Editor has identified that the English language must be improved. PeerJ can provide language editing services - please contact us at [email protected] for pricing (be sure to provide your manuscript number and title). Alternatively, you should make your own arrangements to improve the language quality and provide details in your response letter. – PeerJ Staff

Reviewer 1 ·

Basic reporting

In this manuscript, authors proposed autonomous customer analytics framework, where an oversampling method is presented to handle class imbalance problem of churn recognition datasets in AIOT based industry of telecom. This research work extends the existing research in three major directions. Firstly, only limited studies examine both churn prediction and customer segmentation in telecom sector. Our work overcomes this limitation by providing unified customers analytics framework and connects both these modules for a more accurate solution. Secondly, a mixed data oversampling method is configured to handle the class imbalance problem. Thirdly, for a better understanding of operators, the reasons behind the customer churn are explored for personalized recommendations. Authors suggested to address the following comments and suggestions when preparing the revised version:
= Abstract: section needs to be re-drafted to be self-contained means it has to clearly show the hypothesis, methodology, techniques and tools used, and the results obtained.
= Keywords: Authors suggested to update the keywords by selecting more relevant terms. Keywords play important role in the appearance of the manuscript in scholars search which will give it more hits and more citations.
= Introduction: authors advised to add one more paragraph at the end of the section to show the organization of the rest of the paper.
= What assumptions authors made during this research work? If there is any.
= Authors suggested to go through the following references and they MAY make use of them in updating the introduction and the related work sections:
- Ali Akbar Movassagh, Jafar A. Alzubi, Mehdi Gheisari, Mohamadtaghi Rahimi, Senthil kumar Mohan, Aaqif Afzaal Abbasi, Narjes Nabipour, “Artificial neural networks training algorithm integrating invasive weed optimization with diferential evolutionary model” Journal of Ambient Intelligence Humanized Computing, https://doi.org/10.1007/s12652-020-02623-6.
- Omar A. Alzubi, Jafar A. Alzubi, Ala’ M. Al-Zoubi, Mohammad A. Hassonah, Utku Kose, “An efficient malware detection approach with feature weighting based on Harris Hawks optimization” Cluster Computing Journal, 2021. https://doi.org/10.1007/s10586-021-03459-1.
- Jayaraman Sethuraman, Jafar A. Alzubi, Ramachandran Manikandan, Mehdi Gheisari, and Ambeshwar Kumar, “Eccentric Methodology with Optimization to Unearth Hidden Facts of Search Engine Result Pages” Recent patents on Computer Science, Vol. 12, No. 2, 2019.
- Mehaboob Mujawar, Jafar A. Alzubi “Fundamentals of Mobile Communication” Redshine, 2022 Edition, ISBN: 978-1-4583-0356-1.
= Conclusion: The conclusion should be abstracted so authors need to consider re-drafting it.
= Authors need to confirm that all acronyms are defined before being used for first time.
= Authors need to confirm that all mathematical notations are defined when being used for first time.
= Authors suggested to proofread the manuscript after addressing all comments to avoid any typo, grammatical, and lingual mistakes and errors.

Experimental design

N/A

Validity of the findings

N/A

Additional comments

N/A

Reviewer 2 ·

Basic reporting

This paper presents an autonomous customer analytics framework with an oversampling metho to handle the class imbalance problem of churn recognition datasets in the telecom-based AIOT industry.
The idea is novel and interesting although the paper needs improvement. The literature review can be significantly improved by comparing the latest 2022 and 2023 published works on churn prediction and customer segmentation
.

Experimental design

The algorithm for the proposed framework is required to follow the flow of the architecture .
The proposed modules and its features have to be explained like Auto Machine Learning (AutoML) oversampling method etc
The personalised decision making model and its impact can be elobrated

Validity of the findings

The results can be organised in a better way for enhanced clarity .Reconsider the flow of the paper.Graphs wherever necessary can be plotted for better comparision .
Suggest the future work that can be carried out

---

## Round 0.2 · accepted · Accept

This decision was made by the fact that all the major issues were addressed by the authors. I expect the remaining language and grammatical issues will be properly fixed.

Reviewer 1 ·

Basic reporting

No comment

Experimental design

No comment

Validity of the findings

No comment

Additional comments

Authors revised the manuscript according to reviewers comments and suggestions. Manuscript quality is enhanced and its scientific value increased. However, going through the manuscript one can easily spot few lingual and grammatical issues which needs to be addressed. so, authors suggested to get the manuscript proofread by a fluent English speaker to avoid such issues. Also, authors needs to double check the references style which has to meet with the journal one.

Reviewer 2 ·

Basic reporting

YES all the issues raised were addressed by the authors

Experimental design

YES all the issues raised were addressed by the authors

Validity of the findings

YES all the issues raised were addressed by the authors

Additional comments

YES all the issues raised were addressed by the authors